# Life-Cycle Assessment of Biofortified Productions: The Case of Selenium Potato

Alessandro Scuderi 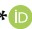, Mariarita Cammarata, Giovanni La Via, Biagio Pecorino and Giuseppe Timpanaro *

Department of Agricultural, Food and Environment (Di3A), University of Catania, via Santa Sofia, 98-100, 95123 Catania, Italy; alessandro.scuderi@unict.it (A.S.); mariar.cammarata@gmail.com (M.C.); giovanni.lavia@unict.it (G.L.V.); pecorino@unict.it (B.P.)
* Correspondence: giuseppe.timpanaro@unict.it

**Abstract:** The increasing micronutrient deficiency within the nutritional habits of the world's population and the growing need for healthy foods have given rise to the development of biofortified crops. In a context where the consumer's attention is focused on a healthy lifestyle and respect for the environment, the cultivation of potatoes enriched with selenium offers an undisputed advantage in the pursuit of this twofold objective. The crop has been analyzed through the life-cycle assessment (LCA) methodology in order to highlight the environmental burden generated by selenium (Se) potato cultivation and to compare it with potato in conventional regime. The LCA highlights how the biofortified product is more sustainable than the conventional one, and this not only provides a benefit for the consumer, but also designates a new time for farmers who have the opportunity to implement more environmentally friendly practices.

**Keywords:** LCA; sustainability; hidden hunger; healthy food; marketing; environment

## 1. Introduction

The micronutrient deficiency typical of many populations' diets, especially in developing countries, is spreading rapidly in different areas of the world. This phenomenon is not simply indicated as "hunger", but as "hidden hunger". Recent FAO (Food and Agriculture Organization) studies show that around two billion people suffer from micronutrient deficiencies, mainly due to poor diets [1]. Micronutrient malnutrition is known as "hidden hunger" because its symptoms have few visible warning signs. Caused by a deficiency of critical micronutrients such as vitamin A, iron (Fe), zinc (Zn), and selenium (Se) hidden hunger impairs the mental and physical development of children and adolescents, thereby generating long-term effects on their livelihoods [2,3]. When people cannot afford to diversify their diets with adequate amounts of fruits, vegetables, or animal-source foods that contain large amounts of micronutrients, deficiencies are inevitable [4]. The enrichment of food through other elements, particularly selenium, allows limiting the onset of widespread diseases such as heart disease, the reduction of immune defenses, male fertility, and hypothyroidism. In addition to nutritional requirements, there are food trends in Western countries in relation to the consumption pattern evolution. The consumer at the time of purchase is driven by interest in the health properties of food, preferring products with a high nutritional content that provide the elements they need. In conjunction, there is a growing demand to find more sustainable products on the market that are made with environmentally friendly practices. To achieve both goals, i.e., to enrich foods with micronutrients and at the same time to obtain a sustainable product, the strategy that can be adopted for some products is the biofortification of crops. This research is based on the environmental impact assessment of the biofortified potato through the application of selenium, and the tool used to quantify emissions is the internationally recognized life-cycle assessment, regulated by ISO 2006 a,b standards [5,6]. Environmental life-cycle analysis (LCA) provides a comprehensive method of analysis that allows comparing alternative

production systems and identifying the greatest consumption of resources and emissions into the environment, highlighting where improvements in techniques are most needed [7].

The aim of the study was to evaluate the sustainability performance of a crop that is subject to a biofortification process, the selenium potato, in a typically suitable area, in order to guide public and private stakeholders in the choice of a strategy that takes due account of the two requirements: favoring a paradigm of green and sustainable agriculture, while respecting the need to support local populations interested in a diet rich in micronutrients

## 2. Materials and Methods

### 2.1. Biofortified Products and the Evolution of Consumption Patterns

Biofortification is the process via which some nutrients are added to food in order to provide a benefit to human health and correct deficiencies in diets [8]. The dilemma related to the definition biofortified food products and the processes used to obtain them derives from the unclear definition proposed in the revision of the "Codex Alimentarius Commission dated 26 November 2018: "Biofortification is any process other than conventional addition to food whereby nutrient content is increased or becomes more bioavailable in all potential food sources for the intended nutritional purposes" [9]. The main objectives that can be achieved through biofortification concern first of all the prevention of risk for human health related to the lack of micronutrients, the correction of dietary habits in some populations, and the improvement of health status; in fact, by enriching food products with antioxidant elements, it is possible to reduce the incidence of cancer and other diseases. In addition to being beneficial for human health, biofortification brings considerable benefits to agricultural productivity, whereby a higher proportion of seedlings survive, initial growth is more rapid, and ultimately yields are higher [10]. Figure 1 shows the practices involved in biofortification and the main effects on reducing micronutrient deficiencies.

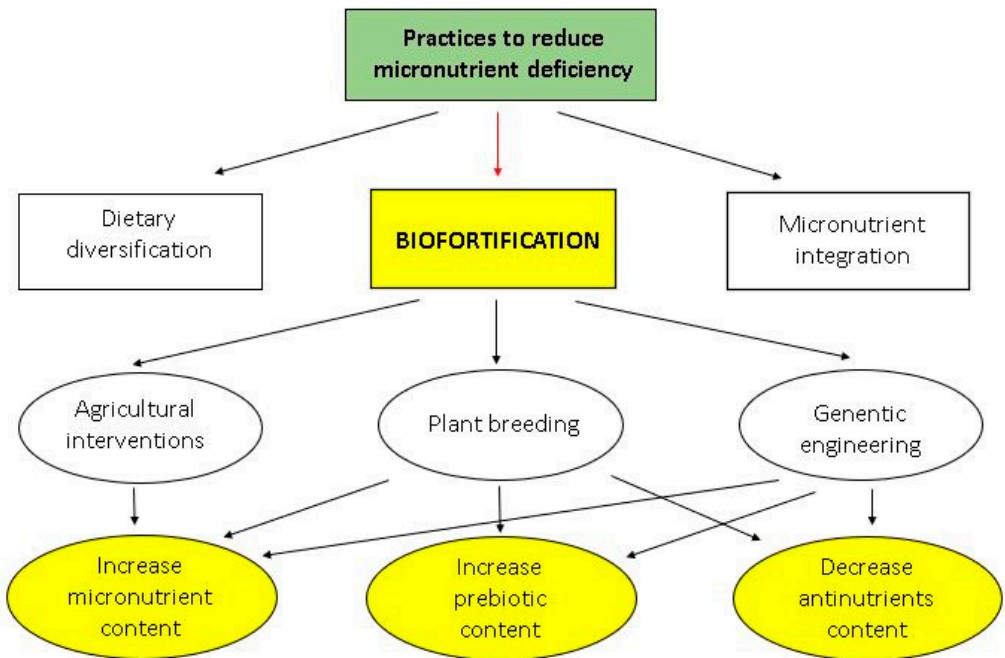

**Figure 1.** Biofortification tools and results.

This research focuses on the cultivation of the selenium-enriched potato (Figure 2) as a response to the needs of Western consumer, modern and health conscious. Selenium is a necessary oligoelement for both humans and animals; it is incorporated as selenocysteine at the active site of a wide range of selenoproteins involved in major metabolic pathways, such as thyroid hormone metabolism, antioxidant defense, and immune function [11]. Low intake of selenium in the diet may cause a number of diseases, including heart diseases,

hypothyroidism, reduced male fertility, weakened immune system, and enhanced susceptibility to infections and cancer [12,13]. Thus, increasing selenium content in food crops offers an effective approach to reduce the selenium deficiency problem in humans and animals [14]. The increase in the content of mineral elements, particularly trace elements such as selenium, in agricultural products, is pursued through the application of both organic and inorganic fertilizers administered directly to the soil or via foliar application. It has been observed that the human population of the world has exceeded the carrying capacity of low-input agriculture, and modern inorganic fertilizers are necessary to obtain the crop yields required to prevent starvation [15]. It is argued, therefore, that the use of inorganic fertilizers must be included in any future strategy for food security. If the widespread use of fertilizers is facilitated, it might be possible to incorporate mineral elements essential for human nutrition before their distribution, as is practiced for selenium in Finland and zinc in Turkey [16]. The use of selenium fertilizers to increase crop selenium concentrations has been particularly successful in both Finland and New Zealand [17–20]. For example, since the incorporation of selenium into all multielement fertilizers used in Finnish agriculture became mandatory in July 1984, selenium concentrations in many indigenous food items have increased over 10-fold [17,18,21,22]. The importance of selenium to human health is of global interest because selenium deficiency occurs to such an extent that many inhabitants of Europe, Australia, New Zealand, India, Bangladesh, and China ingest insufficient amounts of Se (less than 10 mg·day$^{-1}$) in their daily diet [23,24]. The use and diffusion of biofortified products are achieved thanks to the dual acceptance by farmers and, above all, by consumers. Farmers' criteria for changing varieties include food and income security, risk factors balanced with higher revenue through increased production, and production efficiency. Adoption of biofortified crops implies that both producers and consumers accept the change in the final product, having equivalent productivity and characteristics [10].

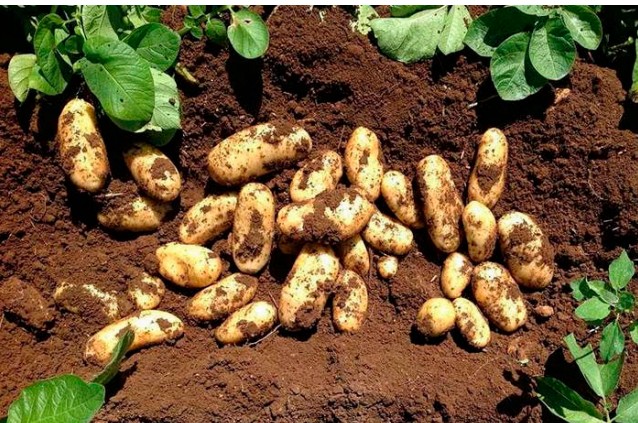

**Figure 2.** Selenium potatoes.

Different studies show that, despite consumers often not being willing to avoid certain foods considered unhealthy, there are increasing changes in consumer behavior that direct purchases toward products considered more beneficial to human health. Today, foods are not only intended to satisfy hunger and to provide necessary nutrients for humans but also to prevent nutrition-related diseases and improve the physical and mental wellbeing of the consumers [25].

The change process we are witnessing is mainly due to more information being available to consumers who have become aware of the connection between a healthy lifestyle and physical wellbeing. There are many scientific studies that have shown over the past decade, with an abundance of experimental data, the close connection between diet and health, particularly in relation to chronic diseases, and have encouraged the development of a growing spectrum of products such as nutraceuticals, medifoods, and vitafoods [26]. It follows that consumers' attention to healthy eating is not exclusively focused on reducing

or eliminating substances considered negative; it tends to move toward the attributes that characterize the product positively, such as freshness and naturalness. It should also be specified that the consumer's choice is driven by the reliability and credibility level that a product transmits at the time of purchase and not by its quality or organoleptic properties; thus, the decision-making power is influenced by the level of knowledge that the consumer has of the product through the information received. The availability of foods enriched with beneficial substances such as antioxidants, vegetable fibers, vitamins, and minerals has, thus, grown under the impetus of a consumer [27] attracted by labels that claim health benefits [28,29]. Information is most likely to be efficient and effective when it manages to meet specific needs of the target audience; hence, it has long been acknowledged that understanding consumers' information-seeking behavior and information processing are crucial to making better marketing decisions [26].

### 2.2. Selenium Potato Life-Cycle Assessment

The research takes into account the cultivation of the selenium potato; for the experimental trials, the production obtained in Sicily in the territory of Syracuse was considered. The choice to carry out the survey on the Syracuse farms is justified because it follows the production techniques. In order to carry out a life-cycle assessment study to evaluate the selenium potato's environmental impact, weekly field surveys were carried out to collect data, and the entrepreneurs' register was used to obtain feedback on the data collected [30]. The selenium potato cultivation, in the area of interest, is comparable to a normal cycle of potato cultivation in conventional agriculture. The difference between the two refers to the application of products containing selenium carried out twice during the final enlargement of fruit and the low impact of product use in the process. The use of selenium causes metabolic functions to stop with consequent yellowing of the plant's epigeal part; therefore, application at the end of the cultivation cycle is preferred. In addition, the studied cultivation differs from conventional ones also in a more careful choice of products used on the crop in order to reduce the environmental impact; in fact, in the case of selenium potatoes, there is not yet a real production standard, but only protocols aimed at obtaining more sustainable production.

Life-cycle assessment is a compilation and evaluation of the inputs, outputs, and the environmental impacts of a product system throughout its life cycle [5]. It takes into account the entire life cycle of a product, ranging from the procurement of raw materials to the disposal of the product itself and possibly its recycling. The method considered allows the identification of improvement opportunities through identifying environmental hot spots in the life cycle of a product, the analysis of the contribution of the life-cycle stages to the overall environmental load, usually with the objective of prioritizing improvements on products or processes, the comparison between products for internal or external communication, and as a basis for environmental product declarations and the basis for standardized metrics and the identification of key performance indicators used in companies for life-cycle management and decision support [31]. LCA methodology is an efficient method to assess impact on the environment (Figure 3), mainly used in industry, as well as in agriculture in recent years [32].

LCA consists of four different and iterative steps defined by the ISO standards: goal and scope definition, inventory analysis, impact assessment, and interpretation [33].

As far as the definition of the goal and scope, the research aimed to identify the main differences between the selenium potato cultivation and conventional potato, in order to define which of the two is characterized by a lower environmental impact and can be considered more sustainable. The life-cycle assessment of the selenium potato was carried out using the SimaPro 9.1 software, within which the recipe midpoint method was chosen. It has a difference in the unit of the indicator for each category. This is because a reference substance has been introduced, such that the characterization factor is a dimensionless number that expresses the strength of an amount of a substance relative to that of the reference substance. For all emission-based impact categories and resource

scarcity, this is a kg reference substance to one specific environmental compartment, while, for land use, it is the area and time integrated for one type of land use [34]. In agricultural product studies, it is the most widely used due to the large number of indicators relating to this topic [35,36]. The definition of the goals and scope includes the identification of the functional unit (FU), the measurement unit to which all inputs and outputs data are related [35]. It was identified as 1 ha of cultivated surface because the objective of the work was to study the cultivation process with its environmental impacts; therefore, the marketable production was of secondary importance. As such, the FU was not identified in 1 kg of product obtained. Important within the first step was also the definition of the system boundaries (Figure 4); the choice fell on a "from cradle to farm gate" analysis, as a consequence of the reasons listed for the functional unit and the lack of data concerning the phases after harvest.

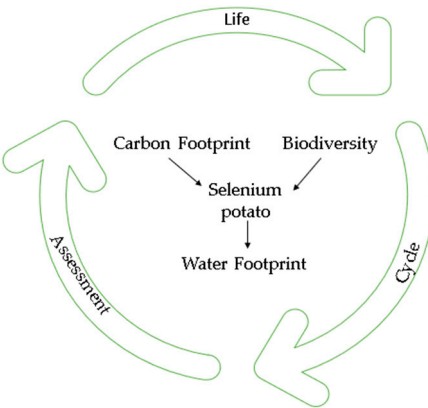

**Figure 3.** Life-cycle assessment (LCA) of selenium potato.

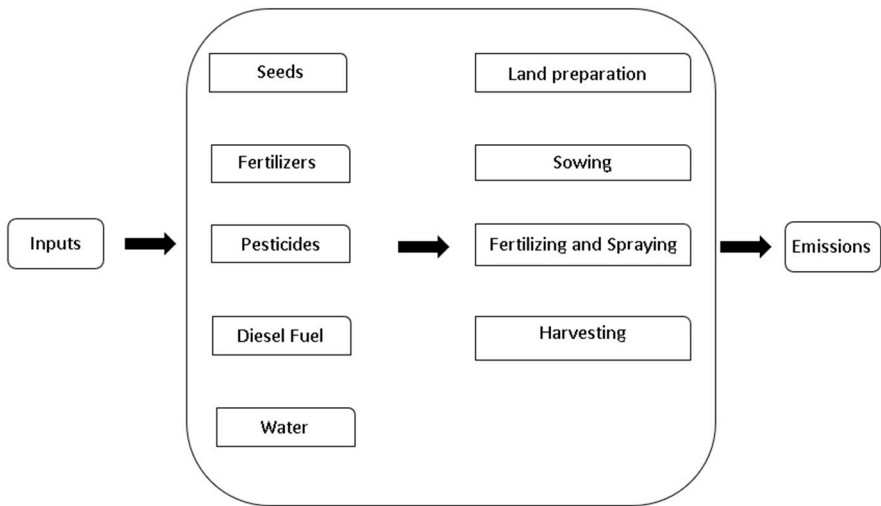

**Figure 4.** System boundaries in the LCA of selenium potatoes.

The second step of an LCA is called inventory analysis; life cycle inventory (LCI) analysis consists of all details about all the environmental inputs (material and energy) and outputs (air, water, and soil emissions) at each stage of the life cycle [37]. Primary data from field surveys carried out with the help of the farmer and secondary data were used for the analysis. The latter refer to fertilizer and fuel production processes. These data were obtained from the Ecoinvent V 3.6 database [38]; in addition, both primary and secondary data were referred to the functional unit. The Ecoinvent database facilitates the application of LCA through the availability of a generic consistent background LCI data system; in fact, the lack of these can have a strong impact on the quality of LCA

studies [39]. The environmental burden generated by the use of machinery and fertilizers was assessed according to Nemecek and Kägi [40]. Concerning the calculation of emissions due to the use of pesticides, the Ecoinvent approach was used, in which the amount of pesticides applied to the crop was considered as emissions to the soil. The substances in the inventories were used as references to correlate the corresponding emissions [40].

The third step is represented by the impact analysis, consisting of "quantifying potential environmental impacts, through the selection of impact categories and, for all of them, relevant indicators and characterization models" [35]. Life-cycle impact assessment (LCIA) helps the interpretation of LCA studies by translating these emissions into environmental impact scores [41]. Characterization factors at the midpoint level are located somewhere along the impact pathway, typically at the point after which the environmental mechanism is identical for all environmental flows assigned to that impact category [34].

The impact categories considered for the evaluation were global warming (GW), stratospheric ozone depletion (SOD), ionizing radiation (IR), ozone formation, human health (OFH), fine particulate matter formation (FP), ozone formation, terrestrial ecosystems (OFT), terrestrial acidification (TA), freshwater eutrophication (FE), marine eutrophication (ME), terrestrial ecotoxicity (TE), freshwater ecotoxicity (F), marine ecotoxicity (M), human carcinogenic toxicity (HCT), human noncarcinogenic toxicity (HNT), land use (LU), mineral resource scarcity (MS), fossil resource scarcity (FS), and water consumption (WC).

In this study, a comparison was also made between the sustainability performance of the selenium potato compared to conventional potatoes in the same areas of study. The data concerning the latter refer to potato cultivation in the same reference area and were taken from an LCA study carried out by Timpanaro et al. [42].

Table 1 shows the measurement units of each impact category analyzed to facilitate their interpretation.

**Table 1.** Impact categories unit of measurement.

| Measurement Units | Definition |
| --- | --- |
| kg CO2 eq | kg carbon dioxide equivalent |
| kg CFC11 eq | kg freon-11 equivalent |
| kBq Co-60 eq | kBq cobalt-60 equivalent |
| kg NOx eq | kg nitrogen oxide equivalent |
| kg $PM_{2.5}$ eq | kg of particulate matter equivalent |
| kg $SO_2$ eq | kg sulfur dioxide equivalent |
| kg P eq | kg phosphorus equivalent |
| kg N eq | kg nitrogen equivalent |
| kg 1,4-DCB | kg 1.4-dichlorobenzene equivalent |
| $m^2$a crop eq | area time (crop) equivalent |
| kg Cu eq | kg copper equivalent |
| kg oil eq | kg oil equivalent |
| $m^3$ | cubic meters |

## 3. Results and Discussion

The results of the LCA (Table 2) show the characterization factors indicating the impact categories analyzed with the relative units of measurement. The total damage was calculated as the sum of the contributions of the processes and materials included in the system boundaries [43]. The results can be a useful starting point to undertake strategies to improve cultivation both in environmental terms and in economic terms to support farmers [44]. The GW category recorded a result of 7584.90 kg $CO_2$ eq, mainly due to the growth phase of the crop in which most of the fertilization, crop protection, and weed control input are concentrated. As the table shows, selenium potatoes have a much lower impact than conventional agriculture, with a relevant percentage difference of −23.73% in this category. Regarding the SOD category expressed in kg CFC11 eq, the value was 0.11, mainly attributable to the growth phase of the crop for the same reasons of the GW category; the selenium potato also in this case reported values much lower than

conventional agriculture with an important difference of −42.77%, which emphasizes the convenience in terms of impact reduction from the analyzed crop. The IR showed a value of 173.72 kBq Co-60 eq, due to the wide use of fertilizers, pesticides, and herbicides during the growing phase, which is the main factor responsible, together with the sowing phase, due to the administration of fertilizers and synthesis products for crop protection [32]. Although no substantial difference with the conventional method was found, there was still a difference of −7.45% to the advantage of biofortified cultivation. Moving on to the OFH category, which describes the impact of this category on human health, expressed in kg NOx eq, it recorded a value of 27.85, which, as the graph shows, as mainly due to the sowing phase and the numerous inputs used in terms of fertilizer and diesel. The selenium potato again showed an important difference from the conventional one, as it had less impact than the latter with a percentage difference of −12.08%, which highlights the importance of the health aspect of the crop in question. FP is the category expressed in kg $PM_{2.5}$, and it had a value equal to 17.11 due both to sowing and to the growth phase of the crop characterized by the administration of the various fertilizers and treatments. The difference with the conventional potato was medium high, −13.26%, in favor of biofortified cultivation, which means that it is more sustainable in this relevant category. In the case of OFT, the selenium potato was −12.02% lower than conventional potatoes, with the percentage showing a high difference in relation to the considered category. Expressed in kg NOx eq, it obtained a value of 28.27, mainly attributable to sowing as a result of the transport process and the various inputs administered to the crop. TA, expressed in kg $SO_2$ eq, had a greater impact on conventional potatoes than on biofortified crops, which obtained a value of 33.29 and a percentage reduction in the impact level of −29.03%. In this case, the percentage indicates a substantial difference and the impact was due to both the sowing and the growth phase for the reasons listed previously. In the FE category, expressed in kg P eq, the selenium potato achieved better results in terms of environmental impact with a very low difference of −0.51% and a value of 1.82 attributable mainly to the sowing phase. Analyzing the ME, in kg N eq, the selenium potato showed a value of 2.27 and a reduction in impact compared to conventional by −2.12%. In this case, the sowing phase was the principal responsible and the difference was low but important in relation to the considered category and impact factors. Concerning TE, the biofortified crop confirmed its sustainability compared to conventional potatoes with a medium high difference of −12.76%; the indicator considered is expressed in 1.4-DCB kg and obtained a score of 51,387.26, largely attributable to sowing. The F category brought a substantial advantage of the biofortified crop over the conventional potato. It is expressed in kg 1.4-DCB, and it had a value of 632, showing an important reduction in impact of −24.32% to the advantage of selenium cultivation. In this case, it was the growth phase that created the greatest impact as a result of the widespread use of synthesis products. Moving on to category M, whose unit of measurement is 1.4-DCB kg, it obtained a value of 665.66 with a reduction in impact compared to the conventional one of −19.60%. This was a substantial difference mainly attributable to the growth phase of the crop. Even in the case of HCT, the selenium potato proved to be more sustainable than the conventional one. Expressed in 1.4-DCB kg, it achieved a score of 179.55 with an important reduction of −12.67% in relation to the considered category, whereby the growth phase also had a greater impact on the environment for the reasons listed above. The latter category analyzed was flanked by HNT with the same unit of measurement as the previous one and a value of 11,660.33 with a high reduction in impact compared to the conventional regime of −16.47%; however, in this case, it was the sowing phase of the crop responsible for the greater impact on human health. At this point, it is possible to analyze the LU category, expressed in $m^2a$ crop eq, which had the greatest impact on the selenium potato due to a greater number of distributions of synthesis products used to control weeds during the sowing phase; the value obtained was 681.77 with a difference of 1.17%. Although the latter was not sufficiently high, it underlined a marginal advantage for conventional cultivation. The MS expressed in kg Cu eq reported a value of 85.13 and a medium high reduction of −5.69%

compared to conventional potatoes, important for the considered category. This was mainly attributable to the growth phase. For FS, on the other hand, the main responsibility was the sowing phase; the category examined, expressed in kg oil eq, obtained a value of 1540.87 and, therefore, had a difference of −17.05, with a decidedly lower impact than conventional potatoes. The last category analyzed is represented by the WC, expressed in $m^3$, which obtained a value of 2489.39 and a marginal difference to the conventional potato with respect to which it obtained an impact reduction of −1.23%. It was more attributable to the growth phase where irrigation interventions were more concentrated. In all the categories analyzed, the impact generated by harvesting was minimal as it did not involve the use of chemical inputs, and only the potato digging machine was used to assist the manual work carried out by farmers.

**Table 2.** Characterization factors and environmental impact per unit of stressor in selenium and conventional potato cultivation (*).

| Impact Category | Unit | P Se | P Conv | Difference % |
|---|---|---|---|---|
| Global warming | kg $CO_2$ eq | 7584.8975 | 9384.7258 | −23.73 |
| Stratospheric ozone depletion | kg CFC11 eq | 0.1058336 | 0.1510982 | −42.77 |
| Ionizing radiation | kBq Co-60 eq | 173.72061 | 186.66713 | −7.45 |
| Ozone formation, human health | kg NOx eq | 27.854544 | 31.219681 | −12.08 |
| Fine particulate matter formation | kg $PM_{2.5}$ eq | 17.107643 | 19.37635 | −13.26 |
| Ozone formation, terrestrial ecosystems | kg NOx eq | 28.274082 | 31.6718 | −12.02 |
| Terrestrial acidification | kg $SO_2$ eq | 33.290309 | 42.953039 | −29.03 |
| Freshwater eutrophication | kg P eq | 1.8246778 | 1.8339521 | −0.51 |
| Marine eutrophication | kg N eq | 2.2722197 | 2.3203019 | −2.12 |
| Terrestrial ecotoxicity | kg 1,4-DCB | 51,387.258 | 57,943.158 | −12.76 |
| Freshwater ecotoxicity | kg 1,4-DCB | 545.14385 | 677.70278 | −24.32 |
| Marine ecotoxicity | kg 1,4-DCB | 665.66013 | 796.10175 | −19.60 |
| Human carcinogenic toxicity | kg 1,4-DCB | 179.5491 | 202.2912 | −12.67 |
| Human noncarcinogenic toxicity | kg 1,4-DCB | 11660.33 | 13,581.233 | −16.47 |
| Land use | $m^2$a crop eq | 681.77309 | 673.79335 | 1.17 |
| Mineral resource scarcity | kg Cu eq | 85.130212 | 89.970775 | −5.69 |
| Fossil resource scarcity | kg oil eq | 1540.868 | 1803.6372 | −17.05 |
| Water consumption | $m^3$ | 2489.3864 | 2519.9408 | −1.23 |

(*) Our elaboration.

Figure 5 highlights what has been said with respect to the most responsible phases of the various categories analyzed.

The study presented some disadvantages related to the chosen crop, such as the limited number of farms analyzed because only they are present in the area considered and the lack of a certified protocol for the cultivation process of the selenium potato, replaced at the moment only by operational indications. In order to exceed these methodological weaknesses, the intention is to implement new research increasing the number of farms under investigation and adopting the protocol of the Consortium "Patata Italiana di Qualità", which is being developed.

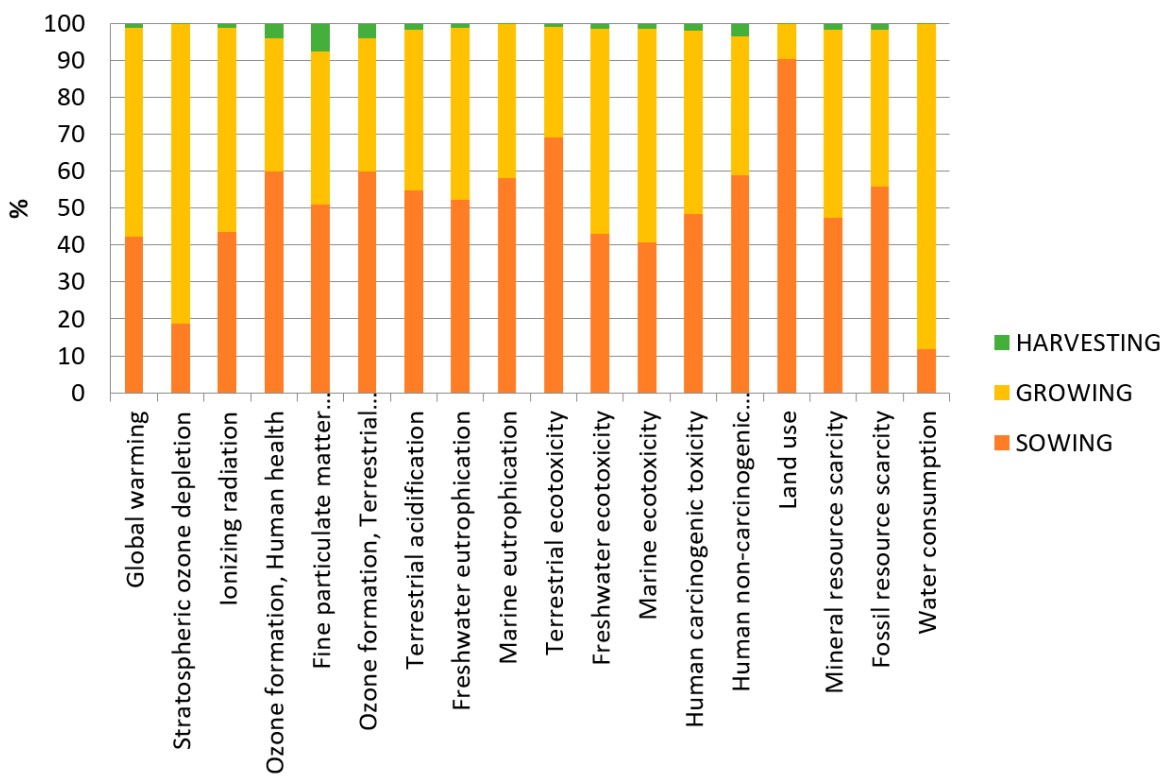

**Figure 5.** Environmental impact of selenium potato cultivation.

## 4. Conclusions

Selenium potato cultivation allows reaching the double objective of reducing the hidden hunger phenomenon and providing the consumer with a product enriched by micronutrients that allows pursuing a healthy lifestyle and helps to lower the risk of various diseases that are very common today [45]. The life-cycle assessment of the crop examined highlights the main cultural operations in which it is possible to intervene in order to reduce the environmental impact and, consequently, to place on the market not only a healthy product but also more sustainable one. The phases in which farmers are invited to pay more attention are the sowing and growing phases, characterized by a wide use of inputs such as fertilizers, chemical products for crop protection and weed control. The life-cycle assessment has highlighted how, despite the applications of selenium-based products and, therefore, a higher number of inputs compared to conventional cultivation, the biofortified crop is more sustainable, i.e., characterized by a lower environmental impact when compared to its conventional counterpart. The most important results were obtained in the categories of global warming, stratospheric ozone depletion, terrestrial acidification, freshwater ecotoxicity, marine ecotoxicity, human carcinogenic toxicity, and human noncarcinogenic toxicity; however, in relation to the importance of the impact categories considered also in other cases, the cultivation of selenium potato brings advantages in terms of environmental sustainability. The main benefits from a social, economic, and environmental point of view make the cultivation of selenium potatoes more sustainable than the conventional method. Overall, the selenium potato as a biofortified product obtained to satisfy the modern health-conscious consumer has shown a significant reduction in input, demonstrating in our case study greater environmental sustainability. The results confirm that "health" products, such as the selenium potato, indirectly have a lower impact. The research indicates that the future of agri-food production will be dictated by respect for the environment and human health independently of the production context. In the future, research will focus, on the one hand, on the analysis of the socioeconomic impacts of selenium potato production and, on the other hand, on the consumer's interest in this

product and the willingness to recognize a price premium for biofortification processes according to well-defined protocols.

This study was based on the intention to highlight the environmental impacts of the cultivation process. The next step will be to identify the environmental impacts related to the product in order to make the consumer aware of them; thus, the functional unit will be 1 kg of selenium potato instead of 1 ha of cultivated area.

**Author Contributions:** The paper is a collaboration of the authors. A.S., M.C., G.L.V., B.P. and G.T. all participated in the design, conceptualisation of the study design, review, editing, formal data analysis, validation and writing of the paper in equal measure. M.C. was also responsible for modelling the data on SIMAPRO. All authors have read and agreed to the published version of the manuscript.

**Funding:** This research received no external funding.

**Data Availability Statement:** Data is contained within the article.

**Conflicts of Interest:** The authors declare no conflict of interest.

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
