# Peer review of "Life-Cycle Assessment of Biofortified Productions: The Case of Selenium Potato"

_asi, doi:10.3390/asi4010001_

Round 1
Reviewer 1 Report
The main topics, that is biofortified crops and their role in our society, is quite interesting and I think it might attract researchers of the field.
As to the main literature, I suggest authors take into account a paper written by Mary Ann Curren (2013) "Life Cycle Assessment: a review of the methodology and its application to sustainability", where the Author highlights 10 aspects of LCA which are often overlooked and explains to what users should pay attention when carrying out an LCA.
In the 2.2 section ("Selenium Potato Life Cycle Assessment"), authors might cite and describe ReCipe midpoint method and the Ecoinvent V 3.6 database. Besides they do not cite any study concerning agricultural products where this method (ReCiPe midpoint) is widely adopted, as they state in the paper.
In the Figure 4, I suggest title is changed deleting the terms "evaluation of", since it deals with the system boundary of the potato LCA.
The inventory analysis is not clearly described or, at least, its description is very concise to be fully understood.
The "Results and Discussion" section in my opinion is focused too much on the observed differences in the environmental performances between conventional and selenium potato and do not comment the observed impact levels in absolute terms in a separate way. Therefore they go into details whithout looking at the findings from an adequate distance.
It is more a "Results" section, since there is not a proper discussion of the results, relating them to similar scientific studies.
There is a misprint, since I noticed a sentence which has been written twice, although not completely. Lines from 250 to 252 should be removed till the full stop after the word "control".
Conclusions drawn in the final section are not supported by results.
Author Response
Review |
Our changes |
In the 2.2 section ("Selenium Potato Life Cycle Assessment"), authors might cite and describe ReCipe midpoint method and the Ecoinvent V 3.6 database. Besides they do not cite any study concerning agricultural products where this method (ReCiPe midpoint) is widely adopted, as they state in the paper. |
Recipe midpoint method was chosen. It has a difference in the unit of the indicator for each category. This is because a reference substance has been introduced, so that the characterization factor is a dimensionless number that expresses the strength of an amount of a substance relative to that of the reference substance. For all emission-based impact categories and resource scarcity, this is a kg reference substance to one specific environmental compartment, while for land use it is the area and time integrated for one type of land use [34]. In agricultural product studies it is the most widely used due to the large number of indicators relating to this topic [35-36]. |
In the Figure 4, I suggest title is changed deleting the terms "evaluation of", since it deals with the system boundary of the potato LCA. |
System boundary in the LCA of elenium potatoes |
The inventory analysis is not clearly described or, at least, its description is very concise to be fully understood. |
Primary data from field surveys carried out with the help of the farmer and secondary data were used for the analysis. The latter refer to fertiliser and fuel production processes. These data were obtained from the Ecoinvent V 3.6 database [38]; in addition, both primary and secondary data were referred to the functional unit. The Ecoinvent database facilitates the application of LCA through the availability of a generic consistent background LCI data system, in fact the lack of these can have a strong impact on the quality of LCA studies [39 |
The "Results and Discussion" section in my opinion is focused too much on the observed differences in the environmental performances between conventional and selenium potato and do not comment the observed impact levels in absolute terms in a separate way. Therefore they go into details whithout looking at the findings from an adequate distance. |
results and discussion have been improved |
There is a misprint, since I noticed a sentence which has been written twice, although not completely. Lines from 250 to 252 should be removed till the full stop after the word "control". |
has been corrected |
Conclusions drawn in the final section are not supported by results |
The most important results were obtained in the categories: Global worming, Stratospheric ozone depletion, Terrestrial acidification, Freshwater ecotoxicity, Marine ecotoxicity, Human carcinogenic toxicity, Human non-carcinogenic toxicity, but in relation to the importance of the impact categories considered also in other cases the cultivation of selenium potato brings its advantage to environmental sustainability. The main benefits from a social, economic and environmental point of view make the cultivation of selenium potatoes more sustainable than the conventional method. Overall, the selenium potato as a bio-fortified product obtained to satisfy the modern health-conscious consumer has shown a significant reduction in input, demonstrating in our case study greater environmental sustainability |
Many thanks for your valuable work and contribution to the improvement of the paper.
The Authors

Reviewer 2 Report
Major comments
The manuscript seems to be a logical and well-defined research with potentially important results. With minor modifications it could have important implications. My suggestions concerns mostly the presentation style, since in some cases it is difficult to understand the implications.
- Introduction and the overall literature review should be improved. There are many related research which are worth to mention (obviously, within the formal limits of the manuscript).
- There are many shortcuts, symbols and notations without sufficient explanation which makes it harder to understand the results. I suggest to put a table in the material and methods section which collects all the shortcuts used in the text, with proper explanations (for example CO2, CFC11, kBq Co-60, NOx etc.). These shortcuts are self-explanatory for researchers but the topic could be valuable for non-experts in this field as well.
- The percentages differences in the results section take value on a high range, from 0.63% up to some 40%. It would be beneficial to distinguish between large and small effects. For example, a 0.5% effect seems marginal for me. Please highlight the importance of these effects in the text.
- Please indicate the potential problems and disadvantages of the research in a few sentences. The part before the Conclusions seems like a good place for this. My concern here is that some of effect (for example the calculated effect on climate change) could be measured very imprecisely, so results are highly indicative in these cases.
Minor comments:
Abstract: the figure is unnecessary in the abstract in my opinion and usually placed in the text.
Line 27: dot is missing after [1]
Line 29: Please provide the missing symbols for iron and zinc.
Figure 1: “BOFORTIFICTAION” should be “BIOFORTIFICATION”
Line 70-75 and line 98-120: This section belongs rather to the introduction than in the Materials and methods section.
Figure 2: Please provide source for the figure.
Line 157: Source is missing for the Ecoinvent V 3.6 database.
Line 250-254: There is some repetition in the text. The sentence in line 252 is basically a rephrasing of the previous sentence.
Line 300: You write: “The results confirm that “health” products indirectly always have a lower impact”. Please lighten this statement. This research proved this statement only for the Se potato, not for all products.
Table 1: this table should contain the difference between Se Potato and Conv. Potato in a comparable manner.
Figure 5: The y-axis contains comma after the numbers, which is unnecessary (“80,” for example). Furthermore, y-axis range should be between 0-100% in my opinion, since no values cross the 100% threshold.

Author Response
There are many shortcuts, symbols and notations without sufficient explanation which makes it harder to understand the results. I suggest to put a table in the material and methods section which collects all the shortcuts used in the text, with proper explanations (for example CO2, CFC11, kBq Co-60, NOx etc.). These shortcuts are self-explanatory for researchers but the topic could be valuable for non-experts in this field as well. |
a table with explanations has been included |
The percentages differences in the results section take value on a high range, from 0.63% up to some 40%. It would be beneficial to distinguish between large and small effects. For example, a 0.5% effect seems marginal for me. Please highlight the importance of these effects in the text. |
the importance of percentages in the text has been indicated |
Please indicate the potential problems and disadvantages of the research in a few sentences. The part before the Conclusions seems like a good place for this. My concern here is that some of effect (for example the calculated effect on climate change) could be measured very imprecisely, so results are highly indicative in these cases. |
The study presents some disadvantages related to the chosen crop, such as the limited number of farms analysed because only they are present in the area considered and the lack of a certified protocol on the cultivation process of the selenium potato replaced at the moment only by operational indications. In order to exceed these methodological weakness the intention is to implement a new research increasing the number of farms under investigation and adopting the protocol of the Consortium "Patata Italiana di Qualità" which is being developed |
Abstract: the figure is unnecessary in the abstract in my opinion and usually placed in the text |
the figure is the graphical abstract |
Line 27: dot is missing after |
has been corrected |
Line 29: Please provide the missing symbols for iron and zinc. |
iron (Fe), zinc (Zn) |
Figure 1: “BOFORTIFICTAION” should be “BIOFORTIFICATION |
has been corrected |
Line 70-75 and line 98-120: This section belongs rather to the introduction than in the Materials and methods section. |
has been corrected |
Figure 2: Please provide source for the figure |
Source: our picture |
Line 157: Source is missing for the Ecoinvent V 3.6 database |
The latter refer to fertiliser and fuel production processes. These data were obtained from the Ecoinvent V 3.6 database [38] |
Line 250-254: There is some repetition in the text. The sentence in line 252 is basically a rephrasing of the previous sentence |
has been corrected |
Line 300: You write: “The results confirm that “health” products indirectly always have a lower impact”. Please lighten this statement. This research proved this statement only for the Se potato, not for all products |
The results confirm that "health" products, such as the Selenium potato, indirectly have a lower impact |
Table 1: this table should contain the difference between Se Potato and Conv. Potato in a comparable manner. |
has been corrected |
Figure 5: The y-axis contains comma after the numbers, which is unnecessary (“80,” for example). Furthermore, y-axis range should be between 0-100% in my opinion, since no values cross the 100% threshold. |
has been corrected |
Many thanks for your valuable work and contribution to the improvement of the paper.
The Authors
